# Serum Metabolomics Uncovers Immune and Lipid Pathway Alterations in Lambs Supplemented with Novel LAB-*Bifidobacterium* Cocktail

**DOI:** 10.3390/ijms26199808

**Published:** 2025-10-09

**Authors:** Roman Wójcik, Angelika Król-Grzymała, Dawid Tobolski, Assel Paritova, Estefanía García-Calvo, Jan Miciński, Grzegorz Zwierzchowski

**Affiliations:** 1Department of Microbiology and Clinical Immunology, Faculty of Veterinary Medicine, University of Warmia and Mazury in Olsztyn, 10-718 Olsztyn, Poland; brandy@uwm.edu.pl; 2Department of Biochemistry, Faculty of Biology and Biotechnology, University of Warmia and Mazury in Olsztyn, 10-719 Olsztyn, Poland; angelika.krol@uwm.edu.pl; 3Department of Large Animal Diseases and Clinic, Institute of Veterinary Medicine, Warsaw University of Life Sciences, 02-787 Warsaw, Poland; dawid_tobolski@sggw.edu.pl; 4Department of Veterinary Sanitation, Faculty of Veterinary and Livestock Technology, S. Seifullin Kazakh Agrotechnical Research University, Astana 010011, Kazakhstan; paritova87@mail.ru; 5Department of Analytical Chemistry, Faculty of Chemical Sciences, Complutense University of Madrid, 28040 Madrid, Spain; egcalvo@ucm.es; 6Department of Sheep and Goat Breeding, Faculty of Animal Bioengineering, University of Warmia and Mazury in Olsztyn, 10-719 Olsztyn, Poland; micinski@uwm.edu.pl

**Keywords:** probiotics, untargeted metabolomics, ruminants, lambs, GC/MS

## Abstract

The ban on antibiotic growth promoters in livestock has intensified the search for effective probiotic alternatives. This study assessed the impact of a novel probiotic cocktail—comprising *Lactobacillus plantarum* AMT14 and AMT4, *L. rhamnosus* AMT15, and *Bifidobacterium animalis* AMT30—on the serum metabolome of lambs using an untargeted GC/MS approach. Sixteen Kamieniec lambs were divided into control and probiotic groups, with serum collected on days 0, 15, and 30. Metabolomic profiling revealed significant alterations in lipid and amino acid metabolism in the probiotic group. By day 15, 38 metabolites were upregulated, including 9,12-octadecadienoic acid, arachidonic acid, and cholesterol. On day 30, key increases included D-glucose, oleic acid, glycine, decanoic acid, and L-leucine. Multivariate analyses (PCA, PLS-DA) demonstrated clear separation between groups, and ROC analysis identified strong biomarkers with high predictive accuracy. These results suggest that probiotic supplementation can beneficially modulate host metabolism, potentially enhancing immune and physiological function in lambs. This highlights the value of multi-strain LAB-*Bifidobacterium* probiotics as a promising strategy for improving health and reducing antibiotic reliance in ruminant production systems.

## 1. Introduction

To fulfill increased energy supply for farm animals, numerous feed additives were introduced in livestock, including antibiotic growth promoters (AGPs) [1]. At that time, it was believed that AGPs used in subtherapeutic doses (below minimum inhibitory concentration [MIC]) will support growth and rearing of farm animals [2] by suppressing the growth of the gut microbiome. The expected outcomes of such practice were reduction in the use of nutrients by microbiome or suppressed production of microbial metabolites. Another expected effect of AGPs was protection of farm animals from infectious pathogens [3]. This was essential, especially since ruminants are agammaglobulinemic; placenta is a very efficient biological barrier for metabolites such as immunoglobulins. Thus, newborn calves, goats, or lambs are susceptible to diarrhea and respiratory infections [4]. Since diseases of the rearing period may also affect future fertility and productivity of animals, lowering the number of diseases during the early period of life of farm animals is the key goal in livestock production [5,6].

However, due to the emerging problem of antibiotic resistance (AR) in bacteria, and increasing consumer awareness, developed countries, including the US and UE, banned AGPs in livestock production. Despite this, the usage of antibiotics in many countries exceeds the suggested global cap set at 50 mg per kilogram of meat production. This means that a large part of the market, mainly low- and middle-income countries, remains without relevant regulations [7,8]. Consequently, alternatives to replace AGPs were searched for, and the usage of probiotics was suggested to help solve the problem.

Lactic acid bacteria (LAB) are probiotics of choice in livestock production because LAB not only modify the gut microbiome but also affect many biochemical pathways of the host, e.g., by vitamin synthesis. Vitamins (biotin, cobalamin, folic acid, nicotinic acid, riboflavin, and thiamine) are essential for LAB survival and are beneficial for the host [9]. Other benefits resulting from the presence of LAB in the host digestive tract include LAB-derived bacteriostatics, which modulate the functions of gut-associated lymphoid tissue (GALT)—a major component of the immune system of the host [10]. Bacteriostatics such as antibiotics, proteases, lysozyme, and organic acids [11,12] support the development of intestinal epithelium and activity of Paneth cells located in intestinal crypts [13]. Because Paneth cells also secrete bacteriostatics, they, along with LAB, antagonize the development of undesirable bacteria strains in the lumen of the gut [14]. Hence, supporting the development and maintenance of the beneficial gut microflora by probiotic supplementation may improve the health of animals, which should result in reduction in antibiotics use in animal husbandry. In a recent study we found that the cocktail of novel LAB strains positively affects the activity of white blood cells [15]. Therefore, the main objective of our experiment was to determine the effect of a mixture of *Lactobacillus plantarum* ATM14 and *Bifidobacterium animalis* ATM30 on the metabolic profile of lambs’ raw blood over a 15- and 30-day period. Analysis of the metabolome of the collected samples was performed using GC/MS.

## 2. Results

Serum metabolomic profiling was performed on 6 CON and 6 supplemented ewes. No differences were observed on d0, confirming that both groups were metabolically comparable prior to supplementation (Appendix A).

### 2.1. Metabolic Alterations After 15 Days of Supplementation

On day 15 of the experiment, GC/MS analysis of blood serum from the control (CON) and probiotic-supplemented (PROBIO) lambs was performed. This analysis allowed to identify 47 metabolites, of which 38 were upregulated and 9 were downregulated (Table 1). To illustrate key differences in metabolite levels between the control (CON) and probiotic (PROBIO) groups, a univariate volcano plot with logarithmic transformation of the data was prepared (Figure 1A). From among 47 identified metabolites, three showed at least a two-fold change in concentration, indicating the most significant differences between the study groups. These key metabolites included 9,12-octadecadienoic acid, cholesterol, and arachidonic acid, all of which were upregulated.

To further analyze the differences in metabolomic profile between the control and probiotic groups, a principal component analysis (PCA) was performed (Figure 1B). The first principal component (PC1) explained 40.9% of the variance, and the second component (PC2) 27%, accounting for 67.9% of the total variance. Control samples were clustered centrally in the graph, while samples from the probiotic group showed greater dispersion. To maximize the separation of the groups observed in PCA, PLS-DA analysis was used (Figure 1C). The graph clearly shows separation between the control and treatment groups, with little scatter within the treatment group. The contribution of the variables was assessed by examining the variable importance in projection score (VIP). Of the 20 most important variables, five were identified as metabolite variables: D-galactose, D-turanose, D-glucose, 9,12-Octadecadienoic acid, and Arachidonic acid (Figure 1D). A ROC test was used to assess the diagnostic potential of these biomarkers (Figure 1E). The area under the curve represents the numerical relationship between specificity (FPR) and sensitivity (TPR) of the metabolite. An AUC of 1.0 indicates excellent predictive accuracy of the diagnostic test.

### 2.2. Metabolic Alterations After 30 Days of Supplementation

After 30 days of the experiment, GC/MS analysis identified 47 metabolites, of which only 5 were downregulated and 42 were upregulated (Table 2). A volcano plot was performed to identify the most variable metabolites between the control (CON) and probiotic-treated (PROBIO) groups. As on day 15, metabolites with significant changes (*p* < 0.05) and high log2-fold change were located in the upper parts of the plot (Figure 2A), suggesting that these metabolites underwent the greatest changes in response to the probiotic. Metabolites that showed significantly higher levels in the PROBIO group included D-glucose, oleic acid, cholesterol, decanoic acid, glycine, hexadecenoic acid, urea, inositol, and L-leucine. Compared to day 15, the list of metabolites expanded. PCA analysis (Figure 2B) showed that the first principal component (PC1) explained 34.6% of the variance, and the second component (PC2) 30.3%, for a total of 64.9% of the total variance. This was a 3% lower result compared to day 15 of the experiment (Figure 1B). In both cases, the control samples clustered in the central part of the graph, indicating a lack of internal differences. However, on day 30, even greater dispersion of the PROBIO group samples was observed. The PLS-DA graph (Figure 2C), similar to day 15 (Figure 1C), showed a clear separation between the control and treatment groups, with even greater dispersion of the PROBIO group samples. The contribution of individual variables was assessed based on the VIP index values. From among 20 variables with the highest VIP values, five were identified as key metabolites: octadecanoic acid, decanoic acid, oleic acid, hexadecenoic acid, and D-myo-inositol (Figure 2D). A ROC test was performed to assess the diagnostic potential of the identified metabolites (Figure 2E). As on day 15 (Figure 1E), AUC values confirmed the high sensitivity and specificity of the selected metabolites in differentiating samples between the control and PROBIO groups, suggesting that these metabolites may play an important role in metabolic differentiation in response to the probiotic.

To contextualize the metabolite changes, we performed over-representation analysis (ORA) separately for day 15 and day 30 using metabolites pre-selected by univariate testing. Panels 3A and 3B present ranked pathways, with enrichment ratios and corresponding *p*-values. Applying a pathway-level significance criterion of unadjusted *p* < 0.05, day 15 showed four significant pathways—Lactose Degradation (largest enrichment ratio), Alpha Linolenic Acid and Linoleic Acid Metabolism, Galactose Metabolism, and Sphingolipid Metabolism—each specific to day 15. Day 30 showed two significant pathways—Fatty Acid Biosynthesis and Arginine and Proline Metabolism—each specific to day 30. No pathway met this threshold at both time points (Figure 3C). The network maps in panel 3C summarize the relatedness among enriched terms.

## 3. Discussion

As research progresses, there is increasing evidence pointing to the crucial role of probiotics in maintaining systemic homeostasis [16]. Recent analyses indicate that gut microbiota can control diverse biological effects by supporting the digestion and absorption of proteins and fatty acids, resulting in modulation of host metabolic pathways [17]. In our experiment, we demonstrated that probiotic supplementation over 15 and 30 days resulted in significant changes in the concentrations of selected metabolites in the serum of lambs. GC/MS analysis revealed a statistically significant increase in blood levels of 9,12-octadecadienoic acid, cholesterol, and arachidonic acid after 15 days of probiotic cocktail supplementation. In lambs, 9,12-octadecadienoic acid (linoleic acid) is converted by the gut microbiota forming arachidonic acid. It was found that feeding animals linoleic acid reduced tumorigenesis of mammary, skin, and colon. Carcinogenesis inhibition mechanism was involved with the reduction in cell proliferation and apoptosis inductions [18]. Arachidonic acid in the cell membrane is metabolized by cyclooxygenases (COX), lipoxygenases (LOX), and cytochrome P450 (CYP) enzymes to many bioactive modulators such as prostanoids and leukotrienes, epoxyeicosatrienoic acids and lipoxins [19]. These compounds influence the release of cytokines and the activity of leukocytes, which plays a key role in the immune defense and inflammatory response of the organism [20,21]. Cholesterol metabolism in lambs is very similar to that in monogastric animals and can be synthesized in the liver and intestines. Blood cholesterol levels are influenced by diet, especially fiber content and probiotics [22].

The obtained results indicate that even 15 days of probiotic supplementation contributes to the modulation of metabolic pathways related to fatty acid synthesis, which may directly have a beneficial effect on the health of lambs [23].

The second part of the experiment examined the effect of long-term (30-day) supplementation with a probiotic cocktail on metabolic changes in the lambs’ serum. A greater increase in metabolite secretion was observed compared to a 15-day supplementation with the same probiotic. GC/MS analysis revealed statistically significant increases in blood concentrations of metabolites such as D-glucose, oleic acid, cholesterol, decanoic acid, glycine, hexadecanoic acid, urea, inositol, and L-leucine. Lipid metabolism is a complex process influenced by many factors. The observed improvement in the lipid profile in lambs, attributed to longer supplementation with the PROBIO cocktail, supports the hypothesis that prolonged probiotic supplementation leads to more pronounced metabolic changes. Oleic acid and linoleic acid are essential components of cell membranes and precursors of bioactive molecules that can influence inflammatory and immune processes [24]. The increased concentration of decanoic acid and the aforementioned fats may result from *Lactobacillus plantarum* and *Bifidobacterium animalis* affecting lipid metabolism pathways such as AMPK/Nrf2 and SREBP-1/FAS, causing fat breakdown, thus preventing adipose tissue accumulation by inhibiting these pathways [25,26]. L-leucine, as a key amino acid, supports immune cell metabolism by increasing CD8+ T cell activity and improving macrophage function [27]. Glycine, one of the amino acids that make up glutathione—a tripeptide with potent antioxidant and immunomodulatory properties—plays a crucial role in regulating the immune response. Furthermore, it is used to alleviate viral infections such as SARS-CoV-2 [28]. Glycine inhibits the production of proinflammatory cytokines such as TNF-α, IL-6, and IL-1β by acting through glycine receptors. In addition, it enhances the expression of the cytokine IL-10 and may act independently of receptors [29].

Probiotic supplementation may increase the expression of glucose transporters in the intestine, leading to higher glucose levels in peripheral blood [30]. Numerous studies on gut health suggest that probiotic supplementation may improve carbohydrate absorption and metabolism by influencing the gut microbiome, which may explain the increase in glucose levels observed in this study [31]. These associations suggest a complex mechanism by which probiotics influence lipid metabolism and the immune response, which may be important for the metabolic health of lambs.

Metabolomics is a rapidly developing field of science, but compared to other areas of “omics,” it remains relatively young. Lack of well-developed databases and stable analytical methods make metabolite identification in complex matrices, such as serum, challenging and limited [32]. Therefore, our results may vary depending on the analytical technique used. It is worth noting that our study has certain limitations. For example, unlike previous studies by Bubnov and colleagues, which demonstrated a reduction in cholesterol following probiotic supplementation [33], our results showed an increase in cholesterol levels. This may be related to the GC/MS method used, which does not distinguish between HDL and LDL fractions. The literature suggests that probiotic supplementation may lower LDL [23,34] and increase HDL [23,35], which may explain the observed increase in total cholesterol in our study.

Additionally, no significant increase in short-chain fatty acid (SCFA) concentrations was observed in the experiment. It should be noted that metabolite analysis was limited to GC/MS, which may compromise the completeness of the metabolomic analysis, especially for SCFA. Due to their rapid metabolism and low blood concentrations, a more precise assessment of SCFAs would require LC/MS [36]. SCFAs are primarily produced in the large intestine during fiber fermentation, and their blood concentrations remain low due to rapid absorption by intestinal cells and metabolism by liver cells, effectively limiting their levels in the circulation [37]. The observed changes in the metabolite profile suggest that long-term probiotic supplementation may not only improve the metabolic health of lambs but also strengthen their immunity. This effect may be important in the context of breeding practices, as increased immunity and improved lipid metabolism can impact the overall condition of the animals, their performance, and the quality of animal products. For this reason, probiotic supplementation could be included as one of the elements of preventive health care in breeding.

This study has several limitations that should be considered when interpreting the findings. First, the per-group sample size was modest. Second, the design included only two post-supplementation time points (15 and 30 days), which precludes modeling temporal trajectories or the durability of effects.

## 4. Materials and Methods

The present study was conducted as part of a broader research project aimed at evaluating the effects of multi-strain probiotic supplementation on the health and development of lambs. Detailed descriptions of the general management, feeding system, and breeding strategy of the flock have been previously described [15,38].

### 4.1. Experimental Design and Sample Collection

The trial was carried out on a flock of Kamieniec sheep maintained at a farm located in Komalwy, Warmia and Mazury region, Poland. Twelve male lambs, born to 3-year-old ewes, were selected for the experiment. At 10 days of age, lambs were divided into two equal groups (n = 6 per group) using the analog method, based on their body weights. The control group (CON) and the experimental group (PROBIO) were comparable in initial body weight. Lambs were housed in two pens with free movement allowed and had continuous access to fresh water free of antibiotics.

Both ewes and lambs received identical diets in accordance with the feeding system adopted at the farm. The ewes had ad libitum access to a total mixed ration (TMR) composed of grass silage (64%), maize silage (32%), concentrate (3.5%), and Milafos L mineral–vitamin premix (0.5%). The concentrate consisted of ground oats (50%), ground wheat (30%), ground maize (10%), and ground soybeans (10%). Mineral licks (Multi-Lisal Se) were available without restriction. All lambs received colostrum within the first hour of life and were exclusively fed maternal milk during the first 10 days postpartum. Starting from day 11 of life, lambs had ad libitum access to concentrate feed.

Group CON lambs were maintained on the basal diet, while group PROBIO lambs received the basal diet supplemented with a multi-strain probiotic preparation. The probiotic cocktail contained four bacterial strains: three *Lactobacillus* isolates (*Lactobacillus plantarum* AMT14, *Lactobacillus plantarum* AMT4, and *Lactobacillus rhamnosus* AMT15) and one *Bifidobacterium animalis* strain (AMT30) (Nature Science, Stawiguda, Poland). The formulation provided a viable count of 1.0 × 10^9^ CFU/g. The probiotic was administered orally as an aqueous suspension once daily according to the following schedule: days 11–20, 10 mL of solution containing 1 g of probiotic/animal; days 21–30, 10 mL containing 2 g of probiotic/animal; and days 31–40, 10 mL containing 3 g of probiotic/animal.

Blood samples were collected from the jugular vein into 9 mL Vacuette^®^ Serum Separation Tubes (Vacuette, Greiner Bio-One, Kremsmünster, Austria) at three time points: prior to probiotic supplementation (day 0) and on days 15 and 30 of the trial. Samples were allowed to clot at room temperature and subsequently centrifuged (10 min at 3000 rpm; MPW 223e centrifuge, MPW Med. Instruments, Warsaw, Poland). The resulting serum was harvested and stored at −20 °C until further analyses.

### 4.2. GC-TOF Compound Identification and Quantification

The extraction and derivatization protocol was adapted from a previously reported method to deproteinize and achieve broad metabolite coverage of polar metabolites in serum [39]. Details of sample preparation, injection of derivatized extracts, quality control (QC), and raw MS data processing have been previously published [40]. Briefly, 100 µL of serum containing 9 µL of internal standard (4-phenylbutyric acid in Acetonitrile (4.5 ppm); Sigma-Aldrich, Bellefonte, PA, USA) was extracted with 800 µL of cold HPLC-grade methanol/HPLC ddH2O water (8:1 vol/vol; Sigma-Aldrich, Bellefonte, PA, USA) and vortexed for 1 min. The samples were kept at 4 °C for 20 min and then centrifuged at 10,000 rpm for 10 min. After centrifugation, 200 µL of the supernatant was transferred to a glass vial insert (250 µL, Agilent, Santa Clara, CA, USA) in a 1.5 mL glass vial with screw cap (Agilent) and evaporated to dryness using a SpeedVac concentrator (Savant Instruments, SDC-100-H, Farmingdale, NY, USA) for 4 h and then using the lyophilizer (LabConco, Kansas City, MO, USA) for 2 h until completely dry.

Extracted metabolites were chemically derivatized prior GC-MS analysis in a two-step procedure: First, 30 μL of 40 mg/mL of methoxyamine hydrochloride in pyridine (Sigma-Aldrich, Bellefonte, PA, USA) was added to each sample and incubated at 37 °C and 500 rpm for 90 min. Secondly, 60 μL of N,O-Bis(trimethylsilyl)trifluoroacetamide (BSTFA; Sigma-Aldrich, Bellefonte, PA, USA) containing 1% (*v*/*v*) of trimethylsilyl chloride (TMCS; Sigma-Aldrich, Bellefonte, PA, USA) was added to each sample and incubated at 60 °C and 500 rpm for 1 h. Samples were finally filtered through 0.22 µm PTFE membranes and transferred into glass inserts for GC-MS analysis. Quality controls (QC) were prepared to ensure the stability and reliability of the metabolomics results. QC samples were obtained by taking 100 μL of supernatant from each previous sample left over, mixing them, and treating the mixture identically to the analytical samples as previously described.

Samples were analyzed by gas chromatography (7890 A Agilent Technologies, Santa Clara, CA, USA) coupled to a time-of-flight (TOF) high-resolution mass spectrometer (GCT premier Micromass, Waters Corp., Milford, MA, USA). An aliquot of 2 μL of each sample was injected in split mode (1:3 ratio) at a temperature of 270 °C. The separation of the metabolites was performed on a ZB-5MS plus column (30 m, 0.25 mm × 0.25 μm, Phenomenex) at a flow ratio of 1 mL/min, using He as gas carrier. The gradient of temperature started at 60 °C and was maintained for 3 min, then it was increased at a rate of 6 °C/min up to 325 °C, and finally maintained during 3 min. The solvent delay was set as 2.5 min, the ion source was an electron ionization (EI) model, and the scan mass range was set between 50 and 800 *m*/*z*. Chromatograms were obtained in total ion current (TIC) mode. The mass spectrometer was tuned and calibrated for mass resolution and mass accuracy on a daily basis using authenticated reference standards. Process coefficients of variation involving instrument performance, chromatography, and mass calibration were checked to ensure quality.

Chromatographic data were analyzed with Mass Lynx software (version 4.2; Waters Corp., Milford, MA, USA), and the peak area of each metabolite was normalized with the corresponding internal standard area and the total protein content of each sample. Metabolites were identified based on both mass spectra and accurate mass using NIST MS search 2.0 library.

### 4.3. Statistical Analysis

Data preprocessing and statistical analyses were performed using the R programming language (version 4.1.0; R Development Core Team, Vienna, Austria, 2008), Python programming language (version 3.7; Rossum & Drake, Python 3 Reference Manual, CreateSpace, Scotts Valley, CA, USA, 2009), JASP software (version 0.19.3; Wageningen, Netherlands) [41], and MetaboAnalyst platform [42].

Missing values were addressed as follows: metabolites with >50% of measurements below the detection limit, or absent in ≥50% of samples, were excluded from the analysis. For the remaining metabolites, missing values were imputed by replacing them with one-half of the minimum positive concentration detected for each metabolite.

Prior to statistical evaluation, data were subjected to log transformation (for univariate analyses) and then mean-centered and scaled by dividing each metabolite value by its standard deviation (for unsupervised and supervised multivariate classification analyses). This normalization procedure was applied to ensure comparability across variables.

The assumptions for univariate analyses—normality (Shapiro–Wilk test) and homogeneity of variance (Levene’s test)—were assessed on both raw and log-transformed data. Results in tables are reported as mean ± standard deviation (SD). Fold change was expressed as the ratio between the PROBIO and CON groups, as well as the log_2_-transformed value of this ratio.

To further investigate the biological roles of the altered metabolites, an enrichment analysis was conducted using the MetaboAnalyst platform. For this purpose, lists comprising all identified metabolites with a *p*-value < 0.1 for both day 15 and day 30 were utilized. A pathway-based Over-Representation Analysis (ORA) was performed against the Small Molecule Pathway Database (SMPDB, accessed on 23 September 2025). The results, including an overview of enriched metabolite sets, network views for each time point, and a Venn diagram illustrating the relationships between metabolite sets, are presented in Figure 3.

## 5. Conclusions

The obtained results suggest that long-term supplementation with a PROBIO cocktail containing *Lactobacillus plantarum* ATM14 and *Bifidobacterium animalis* ATM30 induces more significant metabolic changes, particularly in lipids and amino acids, which may support the metabolic health of lambs. The observed changes in fatty acids, glucose, glycine, and L-leucine levels indicate a positive effect of probiotics on immune and anti-inflammatory pathways. The use of GC/MS, although effective, may limit the detection of some metabolites, such as SCFAs. The results confirm the complexity of the microbiome’s interactions with metabolism and immunity, highlighting the potential of probiotics in animal health management and their possible application in animal husbandry.

## Figures and Tables

**Figure 1 ijms-26-09808-f001:**
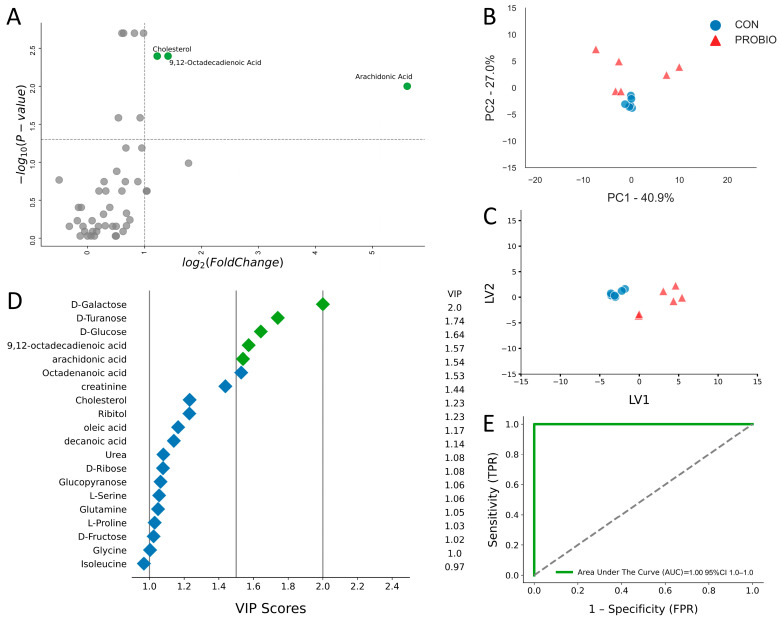
(**A**) Volcano plot showing metabolites with fold changes log_2_(FC) ≥ 1 (green dots) and *p* ≤ 0.05. Gray dots refer to all the other metabolites identified in the dataset whose relative concentrations are not significantly changed between CON and PROBIO group and fold changes log_2_(FC) ≤ 1. (**B**) Principal component analysis (PCA) and (**C**) Partial least squares-discriminant analysis (PLS-DA) of 6 control lambs (CON) and 6 PROBIO lambs at 15 days of the study. (**D**) Variables ranked by variable importance in projection (VIP), (**E**) ROC curve for five top performing metabolites in VIP scores (green diamonds).

**Figure 2 ijms-26-09808-f002:**
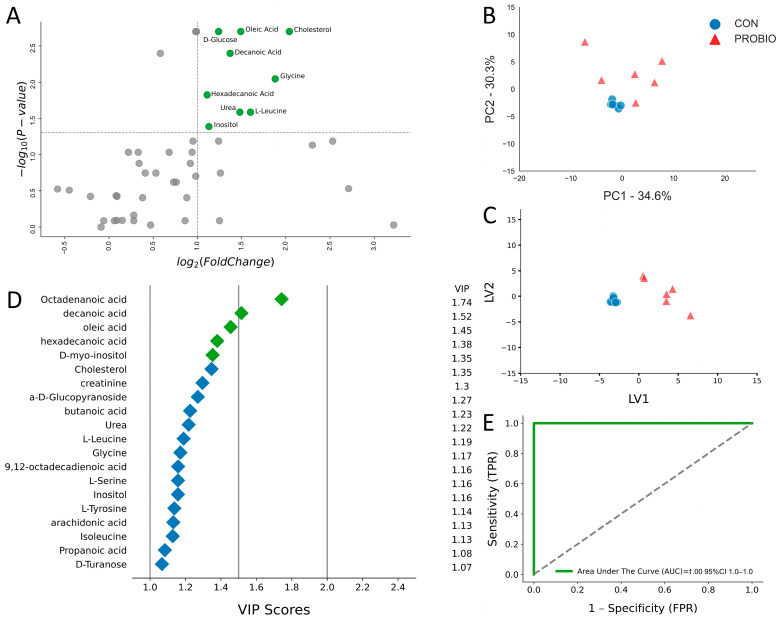
(**A**) Volcano plot showing metabolites with fold changes log_2_(FC) ≥ 1 (green dots) and *p* ≤ 0.05. Gray dots refer to all the other metabolites identified in the dataset whose relative concentrations are not significantly changed between CON and PROBIO group and fold changes log_2_(FC) ≤ 1. (**B**) Principal component analysis (PCA) and (**C**) Partial least squares-discriminant analysis (PLS-DA) of 6 control lambs (CON) and 6 PROBIO lambs at 30 days of study. (**D**) Variables ranked by variable importance in projection (VIP), (**E**) ROC curve for five top performing metabolites in VIP scores (green diamonds).

**Figure 3 ijms-26-09808-f003:**
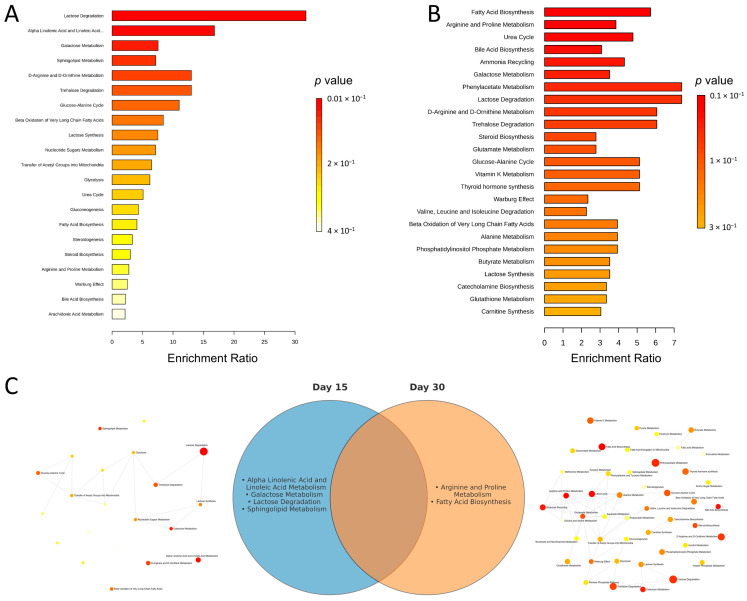
Over-representation analysis (ORA) of serum metabolites on day 15 and day 30. (**A**,**B**) Enriched metabolic pathways ranked by *p* value for day 15 (**A**) and day 30 (**B**); bar length denotes the enrichment ratio, and the color scale encodes the raw *p* value. ORA was performed against SMPDB using metabolites pre-selected by univariate testing. (**C**) Summary of the ORA results. The central Venn diagram lists pathways that met the pathway-level criterion *p* < 0.05:. Left and right network views depict relationships among enriched terms for day 15 and day 30, respectively.

**Table 1 ijms-26-09808-t001:** Concentrations of altered serum metabolites [mean (SEM)] in controls (CON) and supplemented (PROBIO) lambs at d 15 as determined by GC/MS approach.

Metabolite, μM	PROBIO	CON	*p*-Value	Fold Change	log_2_ (FC)	PROBIO/CON
Number of cases	6	6	-	-	-	-
9,12-Octadecadienoic Acid	0.60 (0.30)	0.23 (0.03)	<0.01	2.66	1.41	UP
Acetic Acid	0.07 (0.04)	0.08 (0.05)	0.59	0.88	−0.18	DOWN
Alanine	0.55 (0.16)	0.49 (0.06)	0.82	1.12	0.16	UP
Arachidonic Acid	0.05 (0.03)	< 0.01	0.01	48.76	5.61	UP
B-Alanine	0.02 (0.02)	0.03 (0.01)	0.94	0.91	−0.13	DOWN
Butanedioic Acid	0.03 (0.01)	0.03 (0.01)	0.70	0.95	−0.08	DOWN
Butanoic Acid	0.95 (0.62)	0.68 (0.18)	0.94	1.41	0.49	UP
Cholesterol	7.42 (4.37)	3.19 (0.39)	<0.01	2.33	1.22	UP
Creatinine	0.61 (0.12)	0.42 (0.09)	0.03	1.45	0.54	UP
Decanoic Acid	0.1 (0.05)	0.05 (0.02)	0.03	1.89	0.92	UP
D-Fructose	0.49 (0.32)	0.27 (0.05)	0.18	1.84	0.88	UP
D-Galactose	0.10 (0.01)	0.05 (0.01)	<0.01	1.76	0.82	UP
D-Glucose	8.63 (1.76)	5.57 (0.34)	<0.01	1.55	0.63	UP
Dihydroxybutanoic Acid	0.02 (0.02)	0.02 (0.01)	0.94	1.04	0.06	UP
D-Myo-Inositol	1.84 (1.00)	1.21 (0.13)	0.24	1.52	0.60	UP
D-Ribose	0.18 (0.06)	0.14 (0.03)	0.48	1.22	0.28	UP
D-Turanose	0.64 (0.09)	0.42 (0.05)	<0.01	1.51	0.60	UP
Galactoric Acid	0.08 (0.05)	0.06 (0.02)	0.70	1.36	0.44	UP
Glucopyranose	0.03 (0.03)	0.01 (0.01)	0.10	3.42	1.77	UP
Glucopyranoside	0.02 (0.02)	0.01 (0.00)	0.57	1.67	0.74	UP
Glutamine	0.83 (0.21)	0.78 (0.10)	0.82	1.07	0.09	UP
Glycine	0.74 (0.40)	1.05 (0.32)	0.18	0.70	−0.51	DOWN
Heptadecanoic Acid	0.05 (0.04)	0.04 (0.01)	0.69	1.24	0.31	UP
Hexadecanoic Acid	1.17 (0.85)	0.74 (0.07)	0.18	1.58	0.66	UP
Hexanoic Acid	0.02 (0.01)	0.02 (0.01)	0.70	0.80	−0.32	DOWN
Inositol	0.12 (0.04)	0.10 (0.02)	0.70	1.14	0.19	UP
Isoleucine	1.03 (0.53)	0.67 (0.05)	0.82	1.53	0.62	UP
L-Asparagine	0.03 (0.03)	0.02 (0.01)	0.47	1.60	0.68	UP
L-Leucine	3.79 (2.96)	1.84 (0.36)	0.24	2.06	1.04	UP
L-Methionine	1.35 (0.84)	1.03 (0.14)	0.39	1.31	0.39	UP
L-Ornithine	0.33 (0.25)	0.33 (0.06)	0.94	1.00	0.00	DOWN
L-Proline	0.55 (0.12)	0.45 (0.06)	0.18	1.22	0.29	UP
L-Serine	0.76 (0.18)	0.66 (0.09)	0.24	1.15	0.20	UP
L-Threonine	0.72 (0.20)	0.67 (0.14)	0.94	1.09	0.12	UP
L-Tyrosine	0.46 (0.16)	0.44 (0.05)	0.59	1.06	0.08	UP
Malic Acid	0.05 (0.02)	0.05 (0.02)	0.82	0.96	−0.05	DOWN
Octadenanoic Acid	1.06 (0.30)	0.54 (0.22)	<0.01	1.98	0.98	UP
Oleic Acid	1.13 (0.46)	0.71 (0.27)	0.06	1.59	0.67	UP
Pentanedioic Acid	0.11 (0.08)	0.06 (0.03)	0.24	2.04	1.03	UP
Phosphoric Acid	0.12 (0.08)	0.13 (0.02)	0.39	0.93	−0.11	DOWN
Propanoic Acid	3.50 (2.62)	2.48 (0.45)	0.70	1.41	0.50	UP
Ribitol	0.09 (0.03)	0.07 (0.01)	0.13	1.43	0.51	UP
Serotonin	0.12 (0.09)	0.14 (0.03)	0.39	0.89	−0.16	DOWN
Tryptophan	0.07 (0.07)	0.04 (0.01)	0.69	1.60	0.68	UP
Urea	6.87 (4.13)	3.57 (0.52)	0.06	1.93	0.95	UP
Valine	0.81 (0.52)	0.57 (0.08)	0.94	1.42	0.50	UP
Xylitol	0.07 (0.02)	0.06 (0.02)	0.24	1.25	0.32	UP

**Table 2 ijms-26-09808-t002:** Concentrations of altered serum metabolites [mean (SEM)] in controls (CON) and supplemented (PROBIO) lambs at d 30 as determined by GC/MS approach.

Metabolite, μM	PROBIO	CON	*p*-Value	Fold Change	log_2_ (FC)	PROBIO/CON
Number of cases	6	6	-	-	-	-
9,12-Octadecadienoic Acid	0.41 (0.26)	0.21 (0.09)	0.20	1.97	0.98	UP
Acetic Acid	0.10 (0.10)	0.04 (0.02)	0.82	2.37	1.25	UP
Alanine	4.25 (9.01)	0.46 (0.03)	0.94	9.29	3.22	UP
Arachidonic Acid	0.06 (0.06)	0.01 (0.01)	0.29	6.57	2.71	UP
B-Alanine	0.02 (0.02)	0.02 (0.00)	1.00	0.94	−0.09	DOWN
Butanedioic Acid	0.04 (0.02)	0.02 (0.01)	0.24	1.69	0.76	UP
Butanoic Acid	1.52 (0.80)	0.64 (0.15)	0.06	2.37	1.24	UP
Cholesterol	9.32 (5.37)	2.27 (1.15)	<0.01	4.11	2.04	UP
Creatinine	0.69 (0.18)	0.46 (0.07)	<0.01	1.49	0.58	UP
Decanoic Acid	0.09 (0.03)	0.03 (0.01)	<0.01	2.58	1.37	UP
D-Fructose	0.19 (0.06)	0.20 (0.05)	0.82	0.96	−0.06	DOWN
D-Galactose	0.09 (0.05)	0.05 (0.01)	0.24	1.66	0.73	UP
D-Glucose	12.38 (9.99)	5.26 (0.59)	<0.01	2.35	1.24	UP
Dihydroxybutanoic Acid	0.02 (0.02)	0.02 (0.01)	0.38	0.86	−0.21	DOWN
D-Myo-Inositol	1.91 (0.73)	0.97 (0.09)	<0.01	1.98	0.98	UP
D-Ribose	0.21 (0.10)	0.13 (0.03)	0.09	1.61	0.68	UP
D-Turanose	0.63 (0.21)	0.43 (0.08)	0.18	1.45	0.53	UP
Galactoric Acid	0.09 (0.08)	0.07 (0.02)	0.94	1.38	0.47	UP
Glucopyranose	0.01 (0.01)	0.01 (0.00)	0.37	1.06	0.08	UP
Glucopyranoside	0.01 (0.01)	<0.01	0.07	4.94	2.30	UP
Glutamine	0.83 (0.20)	0.66 (0.11)	0.09	1.26	0.33	UP
Glycine	3.10 (3.84)	0.84 (0.16)	<0.01	3.68	1.88	UP
Heptadecanoic Acid	0.06 (0.06)	0.06 (0.06)	0.81	1.11	0.15	UP
Hexadecanoic Acid	1.92 (0.71)	0.89 (0.30)	0.01	2.16	1.11	UP
Hexanoic Acid	0.03 (0.03)	0.03 (0.02)	0.81	1.06	0.09	UP
Inositol	0.15 (0.09)	0.07 (0.02)	0.04	2.19	1.13	UP
Isoleucine	1.24 (0.66)	0.65 (0.09)	0.09	1.92	0.94	UP
L-Asparagine	0.02 (0.02)	0.02 (0.01)	0.29	0.67	−0.58	DOWN
L-Leucine	5.17 (3.42)	1.71 (0.43)	0.03	3.03	1.60	UP
L-Methionine	1.54 (0.96)	1.18 (0.28)	0.39	1.30	0.38	UP
L-Ornithine	0.23 (0.19)	0.31 (0.09)	0.31	0.73	−0.45	DOWN
L-Proline	0.58 (0.10)	0.50 (0.06)	0.09	1.17	0.22	UP
L-Serine	0.89 (0.23)	0.70 (0.05)	0.13	1.26	0.34	UP
L-Threonine	0.64 (0.24)	0.53 (0.09)	0.82	1.21	0.28	UP
L-Tyrosine	0.65 (0.35)	0.34 (0.08)	0.06	1.93	0.95	UP
Malic Acid	0.08 (0.06)	0.03 (0.01)	0.18	2.39	1.26	UP
Octadenanoic Acid	1.39 (0.24)	0.70 (0.07)	<0.01	1.99	0.99	UP
Oleic Acid	1.93 (0.79)	0.69 (0.16)	<0.01	2.81	1.49	UP
Pentanedioic Acid	0.15 (0.12)	0.08 (0.02)	0.82	1.82	0.86	UP
Phosphoric Acid	0.10 (0.13)	0.09 (0.02)	0.38	1.07	0.09	UP
Propanoic Acid	11.20 (14.29)	1.93 (0.51)	0.06	5.80	2.53	UP
Ribitol	0.11 (0.09)	0.06 (0.01)	0.13	1.90	0.92	UP
Serotonin	0.23 (0.08)	0.17 (0.02)	0.18	1.33	0.41	UP
Tryptophan	0.03 (0.04)	0.03 (0.02)	0.69	1.22	0.28	UP
Urea	10.13 (6.30)	3.64 (0.88)	0.03	2.78	1.48	UP
Valine	0.95 (0.64)	0.52 (0.05)	0.39	1.84	0.88	UP
Xylitol	0.04 (0.02)	0.04 (0.01)	0.82	1.04	0.06	UP

## Data Availability

All relevant data are contained within the manuscript.

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
