# Peer review of "Serum Metabolomics Uncovers Immune and Lipid Pathway Alterations in Lambs Supplemented with Novel LAB-Bifidobacterium Cocktail"

_ijms, 2025, doi:10.3390/ijms26199808_

Round 1

Reviewer 1 Report

Comments and Suggestions for Authors

Roman et al have investigated the impact of prebiotic cocktails on metabolomics of lambs. The subject of the manuscript is worth investigation, and the study is done nicely. There are some comments which should be addressed in order to realise the importance of the study:

  1. The sample size per group is small and this must be acknowledged in the study.
  2. 1 has data from 6 animals. Authors need to mention why it does not have data from all 8 animals.
  3. Have authors done baseline analysis of the two groups for the metabolites? If so, it should be mentioned and if not, then it should be done.
  4. Authors have shown different metabolites to be different in the two groups at the two intervals tested. This is interesting and worth discussing. Authors need to do something like pathway enrichment analysis to understand whether metabolic pathways are active at two time-points.
  5. Title mentions immune pathway alterations but not mentioned/highlighted in results. 
  6. Results section should be divided into sections and have headings for better understanding of the reader.

Author Response

Reviewer 1: Roman et al have investigated the impact of prebiotic cocktails on metabolomics of lambs. The subject of the manuscript is worth investigation, and the study is done nicely. There are some comments which should be addressed in order to realise the importance of the study:

 R1: The sample size per group is small and this must be acknowledged in the study.

AU: We sincerely thank the Reviewer for highlighting the issue of sample size. We agree that this is a key limitation and have now made it explicit in the Discussion (Limitations). We also tempered our conclusions accordingly.

Text added to the manuscript discussion: “This study has several limitations that should be considered when interpreting the findings. First, the per-group sample size was modest. Second, the design included only two post-supplementation time points (15 and 30 days), which precludes modeling temporal trajectories or the durability of effects.”

R1: 1 has data from 6 animals. Authors need to mention why it does not have data from all 8 animals.

AU: We thank the Reviewer for pointing this out. The correct sample size is n = 6 animals per group; the mention of n = 8 was an inadvertent drafting error. We have corrected this number throughout the manuscript (Methods, Results, figure legends, and Supplementary materials) and confirm that all analyses were conducted on the six animals, so the results and conclusions remain unchanged.

R1: Have authors done baseline analysis of the two groups for the metabolites? If so, it should be mentioned and if not, then it should be done.

AU: We agree with the reviewer that baseline analysis is essential We performed a baseline comparison at d0 (CON vs PROBIO). No significant differences were detected after FDR correction. Results are presented as principal component analysis (PCA) of 6 control lambs (CON) cows and 6 PROBIO lambs at 0 day of study. This figure shows no separation of two groups.

R1: Authors have shown different metabolites to be different in the two groups at the two intervals tested. This is interesting and worth discussing. Authors need to do something like pathway enrichment analysis to understand whether metabolic pathways are active at two time-points.

AU: We thank the Reviewer for this valuable suggestion. In response, we performed an over-representation analysis (ORA) separately for Day 15 and Day 30 to contextualize the metabolite-level findings. As detailed in Methods, metabolites were pre-selected within each time point by univariate testing and tested against SMPDB using a hypergeometric framework. Panels 3A and 3B now present the ranked pathway results (ordered by raw p value, with bar length indicating the enrichment ratio and the color scale encoding raw p). A concise synthesis is provided in Panel 3C: using a pathway-level criterion of p < 0.05, Day 15 yielded four significant pathways Lactose Degradation, Alpha Linolenic Acid and Linoleic Acid Metabolism, Galactose Metabolism, and Sphingolipid Metabolism each specific to Day 15; Day 30 yielded two significant pathways Fatty Acid Biosynthesis and Arginine and Proline Metabolism each specific to Day 30. No pathway met this threshold at both time points, indicating time-specific metabolic remodeling. The revised text in Results and the updated Figure 3 address the Reviewer’s request and improve biological interpretability of the time-point differences.

R1: Title mentions immune pathway alterations but not mentioned/highlighted in results. 

AU:  We thank the Reviewer for this helpful observation. The Results highlight immune-relevant pathways identified by ORA (Figure 3): Sphingolipid Metabolism at Day 15 (immune-cell signaling) and Arginine and Proline Metabolism at Day 30 (substrates for nitric-oxide–mediated immune effector functions). These findings justify referring to immune-relevant pathway alterations in the title; therefore, no additional textual changes were introduced.

R1: Results section should be divided into sections and have headings for better understanding of the reader.

AU: Thank you for this helpful suggestion. We have reorganized the Results section into subsections to improve clarity and to facilitate easier reading and understanding of the manuscript.

Reviewer 2 Report

Comments and Suggestions for Authors

The manuscript is well written; however, I have a few questions that should be addressed prior to publication: Where are the IACUC protocols for this study? Why did the authors choose only three time points? Most gut microbiome and Lactobacillus studies include longer time frames to better justify the effects of probiotics on gut microbial populations. Why did the authors not include proteomic profiling of the immune response? Serum proteomics could provide valuable insights for immune profiling. 

Author Response

Reviewer 2: The manuscript is well written; however, I have a few questions that should be addressed prior to publication:

R2: Where are the IACUC protocols for this study?

AU: Thank you for pointing this out. All animal procedures were approved by the Local Ethical Committee for Animal Experimentation in Białystok, Poland (Approval no. 34/2021, May 19th, 2021), in accordance with EU Directive 2010/63/EU. This information is provided in the Institutional Review Board Statement, after Conclusions section.

R:2 Why did the authors choose only three time points? Most gut microbiome and Lactobacillus studies include longer time frames to better justify the effects of probiotics on gut microbial populations.

AU: Our primary objective was to quantify early-to-intermediate systemic metabolic responses to the probiotic cocktail; we did not perform gut microbiome sequencing in this study because the focus was metabolism rather than compositional microbiota shifts. Lactobacillus and Bifidobacterium can rapidly release bioactive cargos—including metabolites (e.g., indole-3-lactic acid) and bacterial/extracellular vesicles (BEVs/EVs)—soon after colonization; these act as inter-kingdom signals and can reach the host circulation, making serum metabolomics an appropriate readout for early effects.

Accordingly, we sampled at day 0, day 15, and day 30 to bracket early and intermediate windows in which short-term probiotic interventions have been shown to produce detectable changes in systemic metabolomes in humans/animals within weeks.

We agree that longer follow-up could further characterize durability and late responses, and we note this as a limitation while outlining extended time-course studies as future work.

R2: Why did the authors not include proteomic profiling of the immune response? Serum proteomics could provide valuable insights for immune profiling. 

AU: We appreciate this important suggestion and fully agree that proteomic profiling could provide valuable complementary insights into host immune modulation. While we would have liked to include such analyses, the scope of this project was constrained by budgetary limitations, and we therefore prioritized metabolomics as a first step.

Round 2

Reviewer 1 Report

Comments and Suggestions for Authors

Authors have revised the mansucript and can be considered for publication.